# Mitigating Hyperglycaemic Oxidative Stress in HepG2 Cells: The Role of *Carica papaya* Leaf and Root Extracts in Promoting Glucose Uptake and Antioxidant Defence

**DOI:** 10.3390/nu16203496

**Published:** 2024-10-15

**Authors:** Mthokozisi Bongani Nxumalo, Nosipho Ntanzi, Hezekiel Mathambo Kumalo, Rene Bernadette Khan

**Affiliations:** Discipline of Medical Biochemistry, School of Laboratory Medicine and Medical Science, College of Health Sciences, University of KwaZulu-Natal, Durban 4000, South Africa; 216016433@stu.ukzn.ac.za (N.N.); kumaloh@ukzn.ac.za (H.M.K.); khanr@ukzn.ac.za (R.B.K.)

**Keywords:** *Carica papaya*, glucose uptake, antioxidants, oxidative stress, HepG2 cells

## Abstract

**Background/Objectives:** Diabetes often goes undiagnosed, with 60% of people in Africa unaware of their condition. Type 2 diabetes mellitus (T2DM) is associated with insulin resistance and is treated with metformin, despite the undesirable side effects. Medicinal plants with therapeutic potential, such as *Carica papaya*, have shown promising anti-diabetic properties. This study explored the role of *C. papaya* leaf and root extracts compared to metformin in reducing hyperglycaemia-induced oxidative stress and their impact on liver function using HepG2 as a reference. **Methods:** The cytotoxicity was assessed through the MTT assay. At the same time, glucose uptake and metabolism (ATP and ∆Ψm) in HepG2 cells treated with *C. papaya* aqueous leaf and root extract were evaluated using a luminometry assay. Additionally, antioxidant properties (SOD2, GPx1, GSH, and Nrf2) were measured using qPCR and Western blot following the detection of MDA, NO, and iNOS, indicators of free radicals. **Results:** The MTT assay showed that *C. papaya* extracts did not exhibit toxicity in HepG2 cells and enhanced glucose uptake compared to the hyperglycaemic control (HGC) and metformin. The glucose levels in *C. papaya*-treated cells increased ATP production (*p* < 0.05), while the ∆Ψm was significantly increased in HGR1000-treated cells (*p* < 0.05). Furthermore, *C. papaya* leaf extract upregulated GPx1 (*p* < 0.05), GSH, and *Nrf2* gene (*p* < 0.05), while SOD2 and Nrf2 proteins were reduced (*p* > 0.05), ultimately lowering ROS (*p* > 0.05). Contrarily, the root extract stimulated SOD2 (*p* > 0.05), GPx1 (*p* < 0.05), and GSH levels (*p* < 0.05), reducing Nrf2 gene and protein expression (*p* < 0.05) and resulting in high MDA levels *(p* < 0.05). Additionally, the extracts elevated NO levels and iNOS expression (*p* < 0.05), suggesting potential RNS activation. **Conclusion:** Taken together, the leaf extract stimulated glucose metabolism and triggered ROS production, producing a strong antioxidant response that was more effective than the root extract and metformin. However, the root extract, particularly at high concentrations, was less effective at neutralising free radicals as it did not stimulate Nrf2 production, but it did maintain elevated levels of SOD2, GSH, and GPx1 antioxidants.

## 1. Introduction

Diabetes mellitus, commonly known as DM, is a widespread chronic disease that poses life-threatening risks and financial complications. It is a metabolic disorder characterised by inadequate insulin secretion and peripheral tissue resistance to insulin’s effect [1]. As a result, it leads to hyperglycaemia, an increase in blood glucose levels due to decreased insulin secretion [1,2]. DM’s high glucose (HG) levels can cause various complications, such as cardiovascular diseases, retinopathy, neuropathy, infertility, and nephropathy [3]. The prevalence of DM is increasing globally, with the International Diabetes Federation (IDF) predicting that approximately 537 million people will be diagnosed with diabetes in 2021. This represents a 12.9% increase in diabetes prevalence, with the highest increase reported in middle-income countries. Future predictions show that these figures will rise to 46% by 2045 [4]. According to a recent study [5], diabetes held the position of the second-most common cause of death in South Africa in 2016 and 2017, while it was the leading cause of death among females. Furthermore, the prevalence of diabetes in South Africa has increased almost 3-fold from 2010 to 2019, reaching a rate of 12.7% in 2019. In 2019, there were an estimated 4.58 million individuals aged between 20 and 79 years living with diabetes in South Africa, with over half (52.4%) remaining undiagnosed.

There are different types of DM, with type 2 diabetes mellitus (T2DM) being the most common, accounting for about 90% of all diabetes cases [1,5]. T2DM is a chronic condition characterised by impaired β-cell function and insulin resistance [5,6]. Risk factors include lack of physical activity, sedentary lifestyle, age, family history, and obesity [7]. Visible symptoms of T2DM manifestation include excessive thirst, frequent urination, dehydration, increased susceptibility to infection, blurred vision, and lethargy [5]. Insulin’s main role is to facilitate the uptake of glucose from the bloodstream to where it is needed for energy or storage [8]. The β-cells in the pancreas produce it and bind to receptors in peripheral tissues, stimulating the use of glucose and storage as lipids and/or glycogen [8,9]. However, T2DM patients experience decreased insulin secretion due to β-cell decline, resulting in reduced insulin levels in the bloodstream [3,10]. Additionally, peripheral tissues become resistant to insulin due to impaired insulin-signalling pathways, leading to high levels of glucose in the bloodstream, or hyperglycaemia [3,11].

Hyperglycaemia in T2DM is associated with increased reactive oxygen species (ROS) and depletion of antioxidants, known as oxidative stress [12]. ROS levels are stimulated by the continuous supply of electron transport chain (ETC) cofactors during ATP production via oxidative phosphorylation in the mitochondria. Superoxide, the initial ROS formed during ETC, leads to the formation of other ROS, such as hydrogen peroxide and hydroxyl radicals. Nitric oxide (NO) and superoxide also produce reactive nitrogen species RNS, such as peroxynitrite. Although, ROS and RNS can be toxic to cell molecules. However, the body has a defence mechanism of antioxidant molecules, including superoxide dismutase 2 (SOD2), catalase (CAT), glutathione (GSH), glutathione peroxidase-1 (GPx1), and vitamins [12,13]. Moreover, Nrf2, recognised as the master regulator, efficiently translocates from the cytoplasm to the nucleus, initiating the antioxidant response to protect against oxidative and nitrative damage [14,15]. Normally, Nrf2 is negatively regulated by Kelch-like ECH-associated protein 1 (Keap1) via its interaction with the Cullin 3 (Cul3) E3 ubiquitin ligase, keeping it under control via proteasomal degradation. However, under oxidative stress, Nrf2 is released from Keap1, translocates to the nucleus, and binds to the antioxidant response element (ARE) of antioxidant and phase II enzyme genes, such as SOD, GPx, and catalase [15]. However, in cases of increased ROS and RNS, antioxidants can be depleted, leading to oxidative stress. Oxidative stress is strongly linked to DM pathogenesis and its complications.

In patients with T2DM, the oxidative stress induced by hyperglycaemia is linked to impaired insulin signalling and subsequent insulin resistance [16]. Prolonged exposure of β-cells to high levels of ROS leads to a decrease in β-cells mass [17], as the β-cells undergo intrinsic apoptosis triggered by high ROS levels, thereby reducing insulin production [16,18]. Furthermore, increased oxidative stress in T2DM is associated with vascular endothelial damage, leading to dysfunction and vascular complications [19]. To stimulate insulin secretion from β-cells, therapeutic drugs, such as metformin, sulfonylureas, and meglitinides, are employed in the treatment of diabetes [3,20]. However, metformin may cause minimal side effects, such as abdominal pain, diarrhoea, and lactic acidosis, while other treatments may lead to weight gain and hypoglycaemia [20,21]. Therefore, there is an urgent need for novel and safe treatments for T2DM that are both effective and affordable.

The World Health Organization (WHO) recognises herbal medicine as a significant traditional medicine used to treat various diseases worldwide [22]. Prior research on medicinal plants is crucial to identify plant toxicity and establish safety for human and animal use [23]. Various medicinal plant extracts contain phytoconstituents with therapeutic benefits globally used to treat diabetes [24]. The potential of South African medicinal plants to treat diabetes is not yet well understood. There is a lack of evidence supporting the traditional use of most plants to manage diabetes, and few studies have investigated their mechanisms of action. Therefore, more research is needed in this area. *Carica papaya*, commonly known as pawpaw, is a plant belonging to the *Caricaceae* family and is grown in tropical countries worldwide [25,26]. Phytochemical screening of pawpaw revealed the presence of alkaloids, flavonols, nicotine, terpinenes, tannins, lycopene, saponins, and other compounds [27]. Pawpaw plant parts are used globally as juice and herbal tea; they are eaten raw or infused to treat various ailments, including stomach ulcers, impotence, malaria, asthma, sinuses, amenorrhea, corns, acne, and high blood pressure [28,29,30]. Additionally, studies have shown that pawpaw has antibacterial, anti-plasmodial, anti-inflammatory, antifertility, antioxidant, antitumor, and anti-diabetic properties [27]. Although some studies have investigated the anti-diabetic properties of pawpaw, there is limited information on its mechanism of action for therapeutic effects [27,31,32,33]. Therefore, this study aims to investigate the effects of *Carica papaya* leaf and root extracts compared to metformin on glucose uptake and reduction of hyperglycemia-induced oxidative stress using HepG2 cells as a model for liver function.

## 2. Materials and Methods

### 2.1. Materials

HepG2 and Hek293 cells were bought from Highveld Biological, located in Johannesburg, South Africa. Whitehead Scientific, also located in Johannesburg, South Africa, supplied all the cell culture reagents and plasticware. Luminometry kits from Promega, based in Madison, WI, USA, and all the antibodies from Cell Signalling Technology (CST), located in Danvers, MA, USA, were purchased from Anatech, located in Johannesburg, South Africa. Bio-Rad, located in Hercules, CA, USA, provided the Western blot reagents, while all other reagents were obtained from Merck based in Darmstadt, Germany, unless specified otherwise.

### 2.2. Cell Culture

The cells that were previously stored were reconstituted with 5 mL of complete culture medium (CCM), which includes Dulbecco’s Modified Eagle Medium (DMED), 1% penicillin-streptomycin-fungizone, 1% L-glutamine, and 10% foetal calf serum, in a 25 cm^3^ sterile flask. This mixture was kept at 37 °C and 5% CO_2_ until the cells reached 80% confluence, and the CCM was changed as necessary. The cells were rinsed with 0.1 M phosphate-buffered saline (PBS) and then trypsinised when they reached about 80% confluence. To obtain the cell count, the trypan blue exclusion method was used.

### 2.3. Carica Papaya Treatment Preparation

The leaves and roots of *Carica papaya* were dried in the dark and crushed, and 10 g was mixed with 200 mL of deionised water. The mixture was stirred continuously for 2 h and then transferred to 50 mL conical tubes. After centrifuging the mixture (2000× *g*, 10 min, at room temperature), the upper aqueous layer was extracted, lyophilised, and stored at 4 °C until needed. The treatment concentrations were prepared by diluting the treatment stock medium with CCM and were utilised in the assays detailed below.

### 2.4. Treatment of Cells

After the cells had adhered to the microtiter plate or reached 80% confluency in the 75 mL flask, the media was removed, and the cells were starved in a serum-free medium for 8 h. Following starvation, hyperglycaemia was induced by incubating the flask/plate of HepG2 and Hek293 cells in HG media for 18 h. The HG media consisted of 25 mM glucose, 10% FCS, 1% pen-strep-fungizone, 1% L-glutamine, and 25 mM HEPES, while low glucose controls were treated with normoglycemic (NG) media consisting of 5 mM glucose, 10% FCS, 1% pen-strep-fungizone, 1% L-glutamine, and 25 mM HEPES. Once sufficient time had elapsed, the media was removed, and the cells were treated with *C. papaya* leaf and root extract. This process was carried out before completing all assays, including the MTT assay.

### 2.5. The 3-(4,5-Dimethylthiazol-2-yl)-2,5-Diphenyltetrazolium Bromide (MTT) Assay

To assess the cytotoxicity of *C. papaya* leaf and root extracts on HepG2 and Hek293 cells, the MTT assay was employed. In a 96-well microtiter plate, a cell suspension of 20,000 cells/well (200 µL/well) was plated in triplicate and left to incubate overnight at 37 °C with 5% CO_2_. Following starvation and induction, various concentrations of *C. papaya* extracts (ranging from 0–3000 µg/mL) were added and incubated for 24 h at 37 °C with 5% CO_2_. After the treatment period, the sample was removed, and 20 µL of 5 mg/mL MTT salt solution (in 0.1 M PBS) and 100 µL of CCM were added. The plate was then incubated for 4 h at 37 °C with 5% CO_2_. Subsequently, the MTT salt solution was replaced with 100 µL of DMSO and incubated for an additional hour at 37 °C with 5% CO_2_. The optical density (OD) was measured at 570/690 nm using a SPECTROstar^®^ Nano microplate reader (BMG LABTECH, Ortenberg, Germany). The absorbance readings were used to calculate the cell viability percentage, and the data were presented graphically as a percentage of cell viability versus *C. papaya* leaf/root treatment concentration.

For Western blotting and qPCR, HepG2 cells were cultured in 75 mL flasks until they reached 80% confluency, while for luminometry, the cells were cultivated in white opaque microtiter plates (with 20,000 cells/well and 200 µL CCM). Following an 8-h starvation period at 37 °C with 5% CO_2_, the cells were acclimated to normal glucose (NG) or HG control conditions for 18 h under the same temperature and CO_2_ levels. Subsequently, they were treated with NG, HG, metformin (1 mg/mL), and *C. papaya* leaf and root extracts (500 and 1000 µg/mL) for 24 h at 37 °C with 5% CO_2_. The treatment medium was retained for the measurement of glucose concentration, MDA, and nitrate concentration. The cells were then prepared for luminometry, Western blotting, and qPCR.

### 2.6. Glucose Quantification and Glucose Uptake Assay

To indirectly determine the amount of glucose that had entered the cells, the concentration of glucose present in the media was obtained using the treatment supernatant from the flask. Specifically, 20 µL of the treatment medium was loaded onto a glucose strip and placed into the ACCU-Chek Active kit glucometer reader (Roche Diagnostics, Basel, Switzerland) to determine the glucose concentration in mmol/L.

To measure glucose uptake in the cells, the Glucose Uptake-Glo™ assay was utilised to detect 2-deoxy-ᴅ-glucose-6-phosphate (2DG6P) [J1341, J1342, and J1343, Promega, Madison, WI, USA]. The cells transport 2-deoxy-ᴅ-glucose (2DG) and use glucose-6-phosphate dehydrogenase (G6PDH) to produce 2-deoxy-ᴅ-glucose-6-phosphate (2DG6P), which reduces NADP+ to NADPH. The cell suspension [20,000 cells/well, (200 µL/well)] was plated into a 96-well microtiter plate in triplicate and incubated overnight (37 °C, 5% CO_2_). After treatment with NG, HG, metformin (1 mg/mL), *C. papaya* leaf (500 and 1000 µg/mL), and *C. papaya* root (500 and 1000 µg/mL) for 24 h at 37 °C and 5% CO_2_, the cells were resuspended with 50 µL of PBS and 25 µL of the prepared 1 mM 2-deoxy-ᴅ-glucose (2DG) was added to each well. The plate was then briefly shaken and incubated for 10 min at room temperature. Subsequently, 25 µL of neutralisation buffer was added, and the plate was briefly placed on a shaker. Then, 100 µL of 2DG6P Detection Reagent (prepared 1 h before use) was added, and the plate was placed on a shaker. After 1 h of incubation, luminescence was recorded using a 0.3–1 s integration on the Turner BioSystems Modulus microplate luminometer (Turner Bio-systems, Sunnyvale, CA, USA), and the results were reported as RLU.

### 2.7. Luminometric Assays

For luminometric assays (ATP, ΔΨm, and GSH/GSSG), the cells were cultured in 25 mL flasks until they reached 80% confluence. Subsequently, the cells were exposed to the appropriate concentration treatment solution (NGC, HGC, HGMet, HGL500, HGL1000, HGR500, and HGR1000; 5000 µL/flask) and then incubated for 24 h at 37 °C with 5% CO_2_. Following the incubation period, the treatment media was removed, and the cells were washed, trypsinised, counted, and seeded in a white opaque 96-well plate (20,000 cells/well in 50 μL 0.1 M PBS). The reagents were prepared according to the manufacturer’s instructions and were added as described below.

#### 2.7.1. Adenosine Triphosphate (ATP) Assay

The ATP activity of HepG2 cells was measured using the Promega Cell Titre-Glo^®^ assay (#G7570, Anatech, Johannesburg, South Africa). The Cell Titre-Glo^®^ ATP reagents were prepared according to the manufacturer’s instructions. Subsequently, 25 μL of the ATP reagent was added to each sample. The plate was then incubated for 30 min at room temperature in the dark. Luminescence was measured using the Turner BioSystems Modulus microplate luminometer (Sunnyvale, CA, USA), and the results were reported as RLU.

#### 2.7.2. Mitochondrial Membrane Potential (ΔΨm) Assay

The JC-10 dye was used to measure ΔΨm, which indicates proper mitochondrial function (MAK159, Sigma-Aldrich, Johannesburg, South Africa). This dye labels normal cells in the mitochondrial matrix, polarizing the mitochondrial membrane and creating red aggregates. However, in cells undergoing cell death, the dye stains the cells green as it moves out of the mitochondria and reverts to its monomeric form [34,35]. A 25 µL JC-10 dye loading solution was added to each well, and the plate was then incubated in the dark at room temperature for 30 min. Subsequently, 25 µL of assay buffer B was added to all wells, and the fluorescence was measured at 540/590 nm and 490/525 nm using a Modulus™ microplate luminometer (Turner Bio-systems, Sunnyvale, CA, USA) in relative fluorescent units (RFU). The red/blue fluorescence ratio intensity was used to determine the ∆Ψm. The data were presented as ∆Ψm (red/blue fluorescence ratio).

#### 2.7.3. Reduced Glutathione (GSH)/Oxidised Glutathione (GSSG) Assay

The concentrations of reduced and oxidised glutathione (GSH/GSSG) within the cells were measured to assess oxidative stress. The intracellular GSH/GSSG levels were determined using the GSH/GSSG-Glo™ assay (Cat. V6611 and V6612). The GSH/GSSG-Glo™ reagents were prepared according to the manufacturer’s instructions, and 25 µL of GSH and GSSG were added to their respective wells. Subsequently, 25 µL of the luciferin generation reagent (LGR) was added to all wells after shaking the plate for 5 min. Following this, the plate was incubated in the dark for 30 min; then, 50 µL of the luminometric detection reagent (LDR) was added, and the plate was equilibrated at room temperature in the dark for 15 min. After incubation, luminescence was measured using a Modulus™ microplate luminometer (Turner Bio-systems, Sunnyvale, CA, USA). The resulting data was reported as mean relative light units (RLU).

### 2.8. Spectrophotometry Assays

For spectrophotometric assays, the HepG2 cells were cultured in 75 mL flasks until they reached 80% confluency. Following an 8-h starvation period at 37 °C with 5% CO_2_, the cells were exposed to normal glucose (NG) or high glucose (HG) control conditions for 18 h under the same temperature and CO_2_ levels. Subsequently, cells were treated with NG, HG, metformin (1 mg/mL), and *C. papaya* leaf and root extracts (500 and 1000 µg/mL) for 24 h at 37 °C with 5% CO_2_. The treatment medium was retained to measure glucose concentration, MDA, and nitrate concentration.

#### 2.8.1. Thiobarbituric Acid Reactive Substances (TBARS) Assay

Lipid peroxidation end-product malondialdehyde (MDA) was evaluated as an indicator of oxidative stress using the TBARS assay. Supernatant samples from the NGC, HGC, HGMet, as well as *C. papaya* treatments (200 µL), along with positive and negative controls, were added to separate test tubes. Then, 200 µL of 7% phosphoric acid (H_3_PO_4_) was added to each test tube, followed by 400 µL of TBA/BHT solution to all tubes except the negative control. The negative control served as the blank, to which 3 mM HCl (200 µL) was added. After vortexing, 200 µL of 1 M HCl was added to all tubes to acidify the pH. The test tubes were then placed in a hot water bath (100 °C) for 15 min. After boiling, the tubes were allowed to cool to room temperature before adding butanol (1500 µL) to each tube and vortexing for 30 s. Samples were then allowed to separate into two distinct layers, and 100 µL of the upper butanol layer was pipetted into a 96-well plate, in triplicate. The absorbance was measured using a SPECTROstar^®^ Nano microplate reader (BMG LABTECH, Ortenberg, Germany) at 532/600 nm. The absorbance readings were used to calculate MDA levels by dividing the absorbance by the absorption coefficient [156 millimolar (mM^−1^)] to generate the mean MDA concentration (μM).
$$[\mathrm{MDA}]=\frac{sample\;absorbance}{{156\;mM}^{-1}}\times 1000$$

#### 2.8.2. Nitric Oxide Synthase (NOS) Assay

The NOS assay was used to quantify nitrates and nitrites in the treatment medium. Standard solutions ranging from 0 to 200 µM were prepared from a 1000 µM sodium nitrate stock solution. Each standard and the treated and untreated supernatant samples (50 µL) were pipetted into wells of a 96-well plate in triplicate. Subsequently, 50 µL VCl3, 25 µL SULF, and 50 µL of NEDD were added sequentially to each well, and the plate was incubated for 45 min in the dark at 37 °C with 5% CO_2_. The absorbance was then measured at 540 nm/690 nm using a SPECTROstar^®^ Nano microplate reader (BMG LABTECH, Ortenberg, Germany). The mean absorbances of sodium nitrate were used to create a standard curve, and the extrapolated equation was utilised to calculate the nitrate and nitrite concentrations of the control and treated samples, which were presented in µM.

### 2.9. Quantitative Polymerase Chain Reaction (qPCR)

After removing the treatment, the flasks were rinsed with 1 M PBS. Then, 500 µL of Triazol and 500 µL PBS were added. The cells were detached from the flask using a cell scraper, and the sample was carefully transferred into 1.5 mL RNAse/DNAse-free microcentrifuge tubes. The tubes were then frozen at −80 °C overnight. The next day, samples were thawed on ice, and 100 µL of chloroform was added to each sample. After vigorous mixing, the samples were incubated at room temperature for 3 min and then centrifuged at 12,000× *g* for 15 min at 4 °C. The aqueous phase was carefully transferred to a new 1.5 mL RNAse/DNAse-free microcentrifuge tube. Isopropanol (250 µL) was added to all samples, and the solution was mixed by gentle flicking before being stored overnight at −80 °C. The following day, samples were thawed on ice and centrifuged at 12,000× *g* for 20 min at 4 °C. The supernatant was removed, and the pellet was resuspended with 75% cold ethanol (500 µL), loosened by flicking, and then centrifuged at 7400× *g* for 15 min at 4 °C. The ethanol was carefully aspirated, and the pellet was air-dried for 1 to 1.5 h at room temperature. After the pellet was completely dry, it was resuspended in 15 µL of nuclease-free water, and the samples were incubated at room temperature for 3 min. After incubation, the samples were mixed and placed on ice in preparation for standardisation using the Nanodrop200 spectrophotometer (ThermoScientific, Waltham, MA, USA). Before adding the samples (1 µL), a blank of nuclease-free water (1 µL) was used, and the concentration and A260/A280 values were recorded into the Nanodrop200 software. The RNA was standardised to 420 ng/mL using nuclease-free water.

A master mix for complementary DNA (cDNA) synthesis was prepared by mixing 5x iScript reaction mix, iScript reverse transcriptase, nuclease-free water, and RNA template. Then, 16 µL of the master mix was added to the qPCR tube, followed by the addition of RNA samples. Next, 4 µL of RNA samples were added to the qPCR tubes and placed in the GeneAmp PCR 97,000 instrument (Applied Biosystems, Waltham, MA, USA). The reaction was conducted at 25 °C for 5 min, 42 °C for 30 min, and 85 °C for 5 min. Finally, 80 µL of nuclease-free water was added to each tube and stored at −80 °C.

To prepare the forward and reverse primer working stocks, 25 µL of the desired primer gene (*Nrf2*) was added separately to two tubes. Then, 75 µL of nuclease-free water was added to each tube. The master mix for each gene contained SYBR green, forward and reverse primers, and nuclease-free water. Subsequently, 11 µL of this mix was added to each well of the PCR 96-well plate in triplicate, and 1 µL of cDNA was included in each well according to the treatments. The plate was then covered with a plastic seal, centrifuged at 2000 rpm for 3 min at room temperature, and placed in the C1000 Touch Thermal Cycler CFX96 Real-Time System (Bio-Rad, Hercules, CA, USA) thermocycler. The incubation process involved initial annealing at 95 °C for 5 min, denaturing at 95 °C for 15 s, annealing at 54 °C for catalase and 55 °C for Nrf2 primers for 4 s, and extension at 72 °C for 2 s for 40 cycles of qPCR standard protocols (Table 1).

The gene of interest was assessed by normalising it against the housekeeping gene, GAPDH, which was amplified concurrently under identical conditions. The Cq values obtained from the melt curve were utilised to compute (2^−ΔΔCT^) Livak and Schmittgen, 2001 [36], enabling the identification of relative changes in mRNA expression (fold change observed in mRNA expression against the HG-control).

### 2.10. Western Blotting

The Western blotting technique was utilised to identify and measure the protein expression in a uniform sample. The crude protein was extracted from both untreated and treated cells using the Cytobuster™ reagent (Novagen, San Diego, CA, USA), which contained phosphatase and protease inhibitors (04906837001 and 05892791001 respectively, Roche (Grenzach, Germany)). In brief, 500 µL of the Cytobuster reagent was added to the 75 mL flasks containing untreated and treated cells and then placed on ice for 30 min and scraped. The resulting cell lysate was transferred into Eppendorf tubes and centrifuged (5 min, 10,000× *g*, 4 °C). The crude protein in the supernatant was collected and quantified using the bicinchoninic acid (BCA) assay.

Bovine serum albumin (BSA) standards (0, 0.2, 0.4, 0.6, 0.8, and 1 mg/mL). A total of 50 µL of standards (ranging from 0 to 1 mg/mL) was prepared and pipetted in triplicate into a 96-well microtiter plate. Afterwards, 50 µL of crude protein samples were added to separate wells in triplicate. Then, 200 µL of BCA solution (4 µL CuSO_4_ and 198 µL BCA) was added to each well, and the plate was incubated at 37 °C for 30 min. Following incubation, the absorbance at 562 nm was measured using a SPECTROstar^®^ Nano microplate reader (BMG LABTECH, Ortenberg, Germany). The average absorbances of the standards were used to create a standard curve, and the derived equation was utilised to determine the protein concentration in each sample. Subsequently, the protein samples were standardised to the lowest concentration of 1 mg/mL. For each protein sample, Laemmli buffer containing 10% sodium dodecyl sulfate (SDS), glycerol, 0.5 M Tris-HCl (pH 6.8), β-mercaptoethanol, 1% bromophenol blue, and deionised water (dH_2_O) was added in a 1:4 ratio. The samples were then boiled at 100 °C for 5 min, allowed to cool to room temperature, and stored at −80 °C until use.

The gels contained a lower 10% resolving gel layer [dH_2_O, 1.5 M Tris-HCl (pH 8.8), 10% (*w*/*v*) SDS, 30% Acrylamide/Bis, 10% ammonium persulfate (APS), and tetramethyl ethylenediamine (TEMED)], and an upper 4% stacking gel layer [dH_2_O, 0.5 M Tris-HCl (pH 6.8), 10% (*w*/*v*) SDS, 30% Acrylamide/bis, 10% APS, and TEMED] was prepared in a Bio-Rad casting stand (Hercules, CA, USA). The standardised protein samples (25 µL) were loaded onto the gels, which were then submerged in 1x running (electrode) buffer (consisting of dH_2_O, Tris, glycine, pH 8.3, at 4 °C). Electrophoresis was conducted using a Bio-Rad compact power supply for 90 min at 150 V while placed on ice.

The gels and nitrocellulose membrane were equilibrated in a cold transfer buffer (containing dH_2_O, Tris, glycine, methanol, pH 8.3) for 10 min. Electro-transfer was performed using the Transblot^®^Turbo^TM^ Transfer system (Bio-Rad, CA, USA) for 30 min (25 V, 2.5 mA). Subsequently, the membranes were blocked for 2 h at room temperature with 2% BSA in Tris-buffered saline (TTBS, 150 mM NaCl, KCl, 25 mM Tris; pH 7.5 and 0.05% Tween 20). The TBBS solution was removed, and the membranes were incubated with the primary antibodies (Table 2) at a 1:1000 dilution in 2% BSA/TTBS for 1 h on a shaker and overnight at 4 °C. The BSA/TTBS was removed, and the membranes were washed with TTBS for 10 min per wash (5 times). Thereafter, the membranes were subsequently probed with a secondary antibody conjugated to horse-radish peroxidase (HRP) [anti-rabbit IgG, 7074], which was diluted with 2% BSA/TTBS (1:2500) in 2% BSA/TTBS for 2 h at room temperature. Subsequently, the nitrocellulose membranes were visualized using the Clarity Western ECL Substrate (catalogue number 1705061, Bio-Rad) and captured using the Chemidoc™ Imaging System and Bio-Rad imaging system (Bio-Rad, CA, USA). Hydrogen peroxide (5%) was used to strip the membranes (37 °C, 30 min), followed by rinsing with TTBS and blocking in 2% BSA in TTBS for 1 h. Afterwards, the membranes were incubated with beta (β)-actin (A0bD12141, Sigma) for 1 h at room temperature (Table 2). After removing β-actin, the membranes were washed three times with TTBS (10 min per wash), and images were captured using the Chemidoc™ Imaging System and Bio-Rad imaging system (Bio-Rad, CA, USA). The band intensity of each protein was measured, and the bands were normalized against β-actin. The data was presented as mean relative band density (RBD).

### 2.11. Data Analysis 

The statistical analysis was performed using GraphPad Prism V5 (GraphPad Software Inc., La Jolla, CA, USA). All experiments were conducted in triplicates and repeated for reproducibility. Statistical significance between treatments and the control was determined using a one-way analysis of variance with a Tukey’s post-test and an unpaired Student’s *t*-test with Welch’s correction (*p* < 0.05). The data was presented as mean ± standard deviation.

## 3. Results

The study was conducted to investigate the effects of *C. papaya* leaf and root extracts on HepG2 cells in both NG and HG media, using metformin (HGMet) as a positive control. The cells were starved for 8 h and then induced with either 5 mmol/L (NG) or 25 mmol/L (HG) glucose concentration for 18 h. After induction, the cells were treated with various concentrations of *C. papaya*. The bar graphs in the figures below show the results for NG untreated cells, HG-control, HGMet (100 µg/mL), and different concentrations of *C. papaya* (HGL500, HGL1000, HGR500, and HGR1000). The hashtag (#) above the bar graphs indicates a significant difference (*p* < 0.05) between NG-control and HGC. At the same time, the asterisk (*) represents HGMet, and *C. papaya* versus HG-control (unpaired *t*-test with Welch’s correction).

### 3.1. Toxicity of C. papaya Leaf and Root Extract on HepG2 Cells

Cell viability of HepG2 cells was tested using the MTT assay to determine the toxicity of *C. papaya* leaf and root extract. Figure 1 shows that cell viability ranged from 85 to 103% compared to the control (100%) for normal glucose control (NGC) and high glucose control (HGC) versus *C. papaya* leaf treatments (0–3000 µg/mL) (Figure 1A,B). The percentage of cell viability showed a similar trend in NG and HG *C*. *papaya* root treatments where the 97–110% range in cell viability was observed (Figure 1C,D).

### 3.2. Toxicity of C. papaya Leaf and Root Extract on Hek293 Cells

The effects of *C. papaya* leaf and root extracts were tested on Hek293 cells to evaluate the impact on kidney function (Figure 2). The viability of both NG, HGC, and HG Hek293 cells treated with *C. papaya* leaf and root extracts at concentrations of 500 and 1000 µg/mL remained at approximately 100%, similar to the NG cells. While HGMet did lead to a slight reduction in cell viability, it remained above 80%.

### 3.3. The Effect of C. papaya Leaf and Root Extracts on Glucose Levels and Uptake in HepG2

#### 3.3.1. Glucose Concentrations in Treatment Medium

To ascertain the quantity of glucose that had been taken up by the cells, the glucose remaining in the treatment media was utilised. The concentration of glucose in the NG-control treatment medium was reduced to minimal levels from an initial 5.1 mmol/L. Conversely, the glucose levels recorded in the HG-control were 7.8 mmol/L. Notably, *C. papaya* leaf and root extract, as well as metformin, were effective in reducing glucose levels from 21.1 mmol/L to low levels after acute exposure (Table 3).

#### 3.3.2. Glucose Uptake in HepG2 Cells

The Glucose Uptake-Glo™ Assay was used to quantify glucose uptake by detecting 2 deoxy-ᴅ-glucose-6-phosphate (2DG6P) levels in cells. The concentration of glucose in the HG-control (584,900 ± 264,200 RLU) cells was significantly (*p* = 0.0048) decreased compared to NG-control (4,401,000 ± 6937 RLU) cells. However, the glucose levels in HGMet-treated cells (4,170,000 ± 1,101,000 RLU, *p* = 0.0870) and HGL500 (3,637,000 ± 1,092,000 RLU, *p* = 0.1129) were non-significantly increased when compared to the HGC (584,900 ± 264,200 RLU) to 7- and 6-fold increase, respectively. Interestingly, HGL1000- (6,530,000 ± 173,400 RLU, *p* = 0.0003), HGR500- (7,001,000 ± 172,100 RLU, *p* = 0.0003), and HGR1000- (5,370,000 ± 177,600 RLU, *p* = 0.0006) treated HG cells showed a significant 11-, 12-, and 9-fold increase in glucose concentration compared to the HGC (Figure 3).

### 3.4. Mitochondrion Functioning and ATP Production

The study evaluated the health and function of mitochondria by measuring the mitochondrial membrane and ATP production using the JC-10 dye and ATP assays. The results showed that the ΔΨm in cells treated with HGC were slightly reduced compared to NGC (Figure 4A). On the other hand, treatments with HGMet, HGL500, and HGR500 showed a non-significant increase of 1.06-, 1.06-, and 1.05-fold in the ΔΨm from 1.212 ± 0.04344 of the control to 1.284 ± 0.09411 (*p* = 0.5561), 1.286 ± 0.009011 (*p* = 0.2342), and 1.276 ± 0.1079 (*p* = 0.6381), respectively. However, HGL1000 induced a 0.93-fold decrease (1.123 ± 0.03725; *p* = 0.2209), while HGR1000 triggered a significant 1.18-fold increase (1.276 ± 0.1079; *p* = 0.6381) compared to HGC (1.212 ± 0.04344).

The study showed that when HG control was implemented, the ATP levels decreased significantly by 0.48-fold to 2,309,000 ± 43,590 (*p* < 0.0001) from the initial levels of 4,835,000 ± 35,460 RLU on the NGC. All treatments significantly increased the ATP concentrations from the HGC (2,309,000 ± 43,590). HGMet resulted in a 2.46-fold increase in ATP concentration, while HGL500, HGL1000, HGR500, and HGR1000 caused increases of 2.81-, 1.62-, 3.01-, and 3.03-fold, respectively. The average RLU values for HGMet, HGL500, HGL1000, HGR500, and HGR1000 were 5,692,000 ± 234,100 (*p* = 0.0049), 6,491,000 ± 98,150 (*p* = 0.0007), 3,731,000 ± 63,830 (*p* = 0.0004), 6,943,000 ± 88,530 (*p* = 0.0020), and 6,943,000 ± 88,530 (*p* = 0.0005) RLU, respectively (Figure 4B).

### 3.5. The Effect of C. papaya on ROS and RNS Production

To indirectly measure the production of ROS, the end product of lipid peroxidation, MDA, was quantified. The concentration of MDA was 1.11-fold higher (*p* = 0.4460) in the HG-control (0.1496 ± 0.01131 µM) compared to the NG-control (0.1346 ± 0.01282 µM) but was reduced to 0.1197 ± 0.0226 µM (*p* = 0.3583) in the HGMet (Figure 5A). Figure 5A indicates a slight increase in the MDA concentration in HepG2 cells treated with *C. papaya*, with MDA levels of 0.1603 ± 0.006410 µM (*p* = 0.4713), 0.1667 ± 0.007402 µM (*p* = 0.0026), and 0.1517 ± 0.002137 µM (*p* = 0.8698) for the HGL500, HGL1000, and HGR500 treatments, respectively. However, the HGR1000 treatment significantly increased MDA levels by 1.53-fold (0.2286 ± 0.005653 µM, *p* = 0.0246).

The NOS assay was utilised to measure the RNS levels of HepG2 cells treated with *C. papaya* to indirectly determine NO concentration through quantifying nitrate/nitrite levels (Figure 5B). The NO concentration was similar under NG-control (15.20 ± 0.05773 µM) and HG-control (14.40 ± 0.5196 µM) conditions but decreased significantly by 0.75-fold in HGMet (12.03 ± 0.1333 µM; *p* = 0.0021) compared to HGC (Figure 5B). However, when compared to HGC, *C. papaya* HGL500 (52.40 ± 1.387 µM, *p* = 0.0014), HGL1000 (84.67 ± 0.6642 µM, *p* < 0.0001), HGR500 (210.8 ± 1.963, *p* = 0.0001), and HGR1000 (406.6 ± 12.00 µM; *p* = 0.0009) triggered a significant increase of 3.45-, 5.57-, 13.87-, and 26.75-fold, respectively (Figure 5B).

### 3.6. The GSH/GSSG Levels

The intracellular concentrations of reduced and oxidised glutathione (GSH/GSSG) were measured to indicate oxidative stress. The presence of hyperglycaemia had a notable impact on GSH levels, decreasing them from 2,352,000 ± 86,530 in NG to 1,566,000 ± 50,200 in hyperglycaemia control cells (*p* = 0.0043). However, treatments proved effective in restoring intracellular GSH levels above the HGC. Metformin (1,752,000 ± 49,500; *p* = 0.0776), HGL1000 (2,113,000 ± 155,200; *p* = 0.0786), and HGR500 (1,666,000 ± 13,120; *p* = 0.1951) were able to increase GSH levels by 1.12-, 1.35-, and 1.06-fold, respectively, under HG conditions (1,566,000 ± 50,200). In addition, treatments with HGL500 (2,327,000 ± 58,320; *p* = 0.0022) and HGR1000 (2,145,000 ± 18,450; *p* = 0.0084) caused a significant 1.49- and 1.37-fold increase in GSH levels (Figure 6A).

The GSH/GSSG ratio was slightly decreased from 3.181 ± 0.02828 in NG-control cells to 2.928 0± 0.2132 (*p* = 0.3615) in HG-control cells. This reduction was similar to the HGMet-treated cells (1.941 ± 0.08717; 0.0503), which reduced the GSSG ratio by 0.66-fold when compared to HGC. However, HGL1000 induced a non-significant 1.26-fold increase (3.680 ± 0.4114; *p* = 0.2030), while a significant elevation of 3.36-fold for HGL500 (9.838 ± 0.05844; *p* = 0.0010), 1.52-fold for HGR500 (4.465 ± 0.1453; *p* = 0.0095), and 4.05-fold for HGR1000 (11.86 ± 0.5038; *p* = 0.0037) in comparison to the HGC (2.928 ± 0.2132), was observed (Figure 6B).

### 3.7. Protein Expression

Protein expression of antioxidant SOD2, GPx1, and iNOS were assessed via Western blot. The iNOS protein expression in NG-control cells (0.3874 ± 0.005828 RBD) was non-significantly increased to 0.4418 ± 0.01912 RBD (*p* = 0.1127) in the HG-control (Figure 7A). However, significant increases in iNOS concentration were noted in the metformin and *C. papaya*-treated cells. In these treatments, iNOS protein expression increased 2.14-fold for the HGMet (0.9465 ± 0.03056 RBD, *p* = 0.0008), 3.14-fold for HGL500 (1.398 ± 0.02878 RBD, *p* = 0.0001), 4.48-fold for HGL1000 (1.980 ± 0.04865 RBD, *p* = 0.0012), 5.54-fold for HGR500 (2.448 ± 0.1022 RBD, *p* = 0.0027), and 3.08-fold for HGR1000 (1.361 ± 0.04552 RBD, *p* = 0.0029) in comparison to the HG-control (0.4418 ± 0.01912 RBD) (Figure 7A).

The expression of SOD2 protein in HG-control cells was reduced by 0.62-fold to 0.4990 ± 0.04088 RBD (*p* = 0.0711) compared to the NG-control (0.6740 ± 0.04896 RBD). Treatment with metformin resulted in a 1.37-fold increase in SOD2 protein expression (0.5740 ± 0.05853 RBD, *p* = 0.3706) compared to HG-control cells. Interestingly, *C. papaya* treatment in hyperglycaemia-induced HepG2 cells led to a decrease in SOD2 protein expression in HGL500 (0.2755 ± 0.02706 RBD, *p* = 0.0198) and HGL1000 (0.3486 ± 0.004017 RBD, *p* = 0.0672) by 0.53 and 0.83, respectively. However, SOD2 expression remained slightly higher than the HGC in HGR500- (0.4723 ± 0.02423; *p* = 0.6142) and HGR1000- (0.4979 ± 0.01391; *p* = 0.9832) treated cells with a 1.10- and 1.19-fold change, respectively (Figure 7B).

The data generated for GPx1 protein expression demonstrated that HGC decreased protein expression from 1.196 ± 0.06501 in the NGC to 0.3051 ± 0.02417 in the HGC (Figure 7C). However, protein expression of GPx1 increased in all HG treatments compared to the HGC cells. Metformin treatment significantly increased GPx1 protein expression to 2.60-fold higher (0.7922 ± 0.01732 RBD, *p* = 0.0005) than the HG-control cells (0.3051 ± 0.02417 RBD). Interestingly, *C. papaya*-treated hyperglycaemia-induced HepG2 cells significantly increased GPx1 protein expression in the HGL500- (0.7193 ± 0.03390 RBD, *p* = 0.0022), HGL1000- (0.6197 ± 0.02540 RBD, *p* = 0.0029), HGR500- (1.016 ± 0.005562 RBD, *p* = 0.0012), and HGR1000- (0.9217 ± 0.02123 RBD, *p* = 0.0003) treated cells by 2.36, 2.03, 3.33, and 3.02, respectively (Figure 7C).

### 3.8. Nrf2 Gene and Protein Expression

The data collected on *Nrf2* gene expression revealed that hyperglycaemia led to a decrease in Nrf2 gene expression from 1.000 ± 0.0003 in the NG-control to 0.3887 ± 0.02756 in the HG-control (*p* = 0.0020). However, *Nrf2* gene expression was modulated by HGMet, resulting in an upregulation of gene expression (0.4470 ± 0.02372; *p* = 0.2071) in comparison to the HG-control. Additionally, *C. papaya* treatments stimulated *Nrf2*, leading to an upregulation of gene expression in HGL500 (0.7132 ± 0.04293; *p* = 0.0993) and a 2.32-fold significant increase in HGL1000 (0.9024 ± 0.03219, *p* = 0.0173) in Figure 8A. However, *C. papaya* root treatments resulted in a non-significant 0.97- and 0.64-fold reduction in HGR500 (0.3787 ± 0.0152; *p* = 0.7712) and HGR1000 (0.1992 ± 0.002403; *p* = 0.0207), respectively (Figure 8A). Also, hyperglycemia significantly increased Nrf2 protein expression from 0.2955 ± 0.008108 in normal glucose to 0.5473 ± 0.002206 in hyperglycemia control cells (*p* = 0.0011). However, HGMet (0.2451 ± 0.02120 RBD; *p* = 0.0049) led to a significant reduction in Nrf2 protein expression with a 0.45-fold change, while HGL500 (0.3930 ± 0.05466 RBD; *p* = 0.1061) and HGL1000 (0.5124 ± 0.02253 RBD; *p* = 0.2626) decreased it by 0.72- and 0.94-fold, respectively, under hyperglycemic conditions (0.5473 ± 0.002206 RBD). Additionally, treatment with HGR500 (0.6732 ± 0.01938 RBD; *p* = 0.0232) caused a significant 1.23-fold increase, and HGR1000 (2,145,000 ± 18,450 RBD; *p* = 0.0084) caused a significant 0.83-fold decrease (Figure 8B).

## 4. Discussion

The International Diabetes Federation’s 2019 report found that 12.8% of South African adults have diabetes mellitus [37]. Many cases of diabetes in Africa go undiagnosed, with 60% of people unaware of their condition. Diabetes can cause high blood sugar and lead to health complications, including damage to vital organs [5]. Metformin and physical activity are usually recommended as the first line of treatment for T2DM, but they can have side effects [3,38]. *Carica papaya*, a medicinal plant, is being studied for its potential anti-diabetic properties [30]. The current study explores the role of *C. papaya* leaf and root extracts compared to metformin in reducing hyperglycaemia-induced oxidative stress and their impact on liver function using HepG2 as a reference.

The HepG2 cells are preferred over primary human hepatocytes for in vitro studies because of their liver-like functions and ability to assess toxicity at various glucose levels [39,40,41]. The liver plays a crucial role in maintaining glucose levels in the bloodstream. The imbalance between glucose uptake and production by the liver can lead to the development of T2DM [10,42,43]. This investigation evaluated the cytotoxicity of *Carica papaya* leaf and root extracts in HepG2 cells. The findings demonstrate that *C. papaya* displays minimal cytotoxicity under normal and HG conditions, with cell viability ranging from 85% to 103% across treatments from 0 to 3000 µg/mL (Figure 1). This suggests that *C. papaya* extract does not negatively impact HepG2 cell viability, aligning with the International Organization for Standardization (ISO) standard of 75% [44]. Additionally, these results are consistent with previous studies that have shown the potential toxicity of *C. papaya* in HepG2 cells within the concentration range of 0.63–20 mg/mL [45,46]. However, our study found that at concentrations of 500 µg/mL and 1000 µg/mL, as well as with metformin (100 µg/mL), cell viability remained at 80–101%. This confirms that these treatments were non-toxic, a conclusion further supported by the cell viability of Human Embryonic Kidney (Hek293) cells (Figure 2). These findings are significant in the context of *C. papaya*’s potential as an anti-diabetic treatment, especially considering the increased risk of diabetic nephropathy (DN) in patients with T2DM, which is the leading cause of end-stage renal disease [5,47].

Based on in vivo reports, extracts from *C. papaya* leaves have anti-diabetic properties. These extracts can reduce glucose levels in circulation by normalising blood glucose levels in diabetic rats [48,49]. To confirm this, our in vitro study assessed glucose concentrations in HG HepG2 cells treated with *C. papaya* leaf, root extracts, and metformin. The results demonstrated that these treatments effectively reduced glucose levels to baseline values (Table 3). These results suggest that *C. papaya* leaf and root extracts, similar to metformin, enhance glucose absorption in HepG2 cells, effectively lowering elevated glucose levels in the bloodstream. However, reduced glucose levels in circulation may indicate increased intracellular glucose accumulation [50]. Therefore, 2-deoxy-ᴅ-glucose (2DG), the analogue of glucose, taken up into cells [51], was quantified by detecting phosphorylated 2-deoxy-ᴅ-glucose-6-phosphate (2DG6P) to determine the rate of glucose uptake into the cells. The results indicated that all *C. papaya* extracts increased glucose uptake compared to HGC, with *C. papaya* HGL1000, HGR500, and HR1000 more effective than HGMet (Figure 3). These findings indicate that *C. papaya* significantly enhances glucose uptake under HG conditions, consistent with the observed reduction in glucose levels in the treatment media (Table 3). This aligns with the in vivo studies that showed *C. papaya* leaf’s ability to reduce hyperglycaemia in diabetic rats [49,52,53]. This effect is attributed to phytochemicals present in *C. papaya*, especially quercetin, transferulic acid, and kaempferol, which have a strong binding affinity for IRS-2 and GLUT-2 proteins, consequently promoting glucose uptake [48,54,55]. However, it is important to note that the accumulation of glucose within cells can be toxic if not used properly for metabolic processes [56]. One of the essential uses of glucose within cells is energy production.

Mammalian cells produce energy by breaking down glucose into pyruvate and 2 ATP in the cytosol via glycolysis [57]. Pyruvate is converted into acetyl-CoA in the mitochondria, which enters the tricarboxylic acid cycle and produces GTP, NADH, and FADH2, which serve as electron carriers for ATP production in the ETC through oxidative phosphorylation [57,58,59]. Consequently, an elevation in intracellular ATP levels can indicate glucose uptake and breakdown. In line with this, our study demonstrated that all *C. papaya* treatments significantly elevated intracellular ATP levels under HG conditions compared to HGC, similar to the effects of HGMet (Figure 4B). This finding implies that *C. papaya* extracts, like metformin, may effectively manage hyperglycemia by improving cellular glucose utilisation and energy production in the mitochondria (Figure 3 and Figure 4B, respectively). The ETC generates the mitochondrial membrane potential (ΔΨm) through the proton pumps of complexes I, III, and IV, which is associated with ATP production [59].

The ∆Ψm is a key indicator of mitochondrial function and health [60,61]. A change in ΔΨm can signal potential mitochondrial dysfunction or impaired energy production, which may be detrimental to cellular health. The ∆Ψm remained unchanged in the HGL500, HGL1000, and HGR500 *C. papaya* treatments compared to HGC, similar to the HGMet treatment, while it significantly increased at HGR1000 (Figure 4A). Therefore, the ETC was functioning properly in HG cells treated with *C. papaya* and metformin, except for HGR1000, where the ΔΨm was increased. This increase in ΔΨm can compromise the mitochondria and lead to its dysfunction, thus reducing ETC activity and ATP production [62,63]. Except for HGR100, this study noted that the mitochondria responded appropriately to elevated intracellular glucose levels. This was indicated by an increase in ATP (Figure 4B) and non-significant alteration in ΔΨm (Figure 4A).

Elevated glucose levels can give rise to free radicals, which are activated by the consistent provision of ETC cofactors during the production of ATP through oxidative phosphorylation in the mitochondria [12]. These radicals are essential for gene expression and signalling. However, an excess of glucose results in an overabundance of these radicals, leading to oxidative stress, which harms cellular components and contributes to various health concerns [59,64]. Malondialdehyde is a highly reactive aldehyde produced as a byproduct of lipid peroxidation and used as a marker of oxidative stress [59]. According to the study, it was observed that MDA concentration remained the same in all *C. papaya* treatments with that of the HGC, except for the HGR1000 (Figure 5A). However, HGMet treatment was found to non-significantly decrease MDA levels in HepG2 cells. In HepG2 cells treated with *C. papaya* under HG levels, the results suggest that damage to lipids caused by ROS was minimised, thus protecting the cells. However, treatment with HGR1000 was found to elevate MDA levels significantly (Figure 5A) due to increased ∆Ψm (Figure 4A), indicating that a high concentration of *C. papaya* root extract might lead to an increased production of ROS.

The formation of ROS occurs when superoxide is released into the mitochondrial matrix due to electron leakage from complexes I and II of the ETC [65]. Superoxide in cells is converted to a less harmful molecule, namely hydrogen peroxide, by the SOD2 enzyme. In this study, *C. papaya* leaf extract decreased SOD2 enzymes, while root extract had a similar effect to HGC (Figure 7B). The SOD2 levels in the cells treated with the *C. papaya* leaf extract may have decreased because of catalysing the formation of hydrogen peroxide. As a result, this suggests that the cells were subsequently compromised and could be vulnerable to damage from ROS unless detoxified through other means. Nonetheless, the *C. papaya* root extract was able to maintain SOD2 availability like metformin above the HGC (Figure 8B), indicating the possibility of detoxification of superoxide to hydrogen peroxide.

Hydrogen peroxide can still cause significant and irreparable damage to cells, especially in the presence of heavy metal derivatives like iron and copper [66]. This is because hydrogen peroxide can promote the production of ROS, such as the hydroxyl radical, a strong oxidant that damages cellular molecules, including lipids, and particularly polyunsaturated fatty acids [66,67]. The MDA levels suggested that ROS caused minimal damage, except for the HGR1000 treatment (Figure 5A). Cells activate defence mechanisms to reduce free radical levels when they increase [59]. It’s important to note that glutathione can counteract hydrogen peroxide by utilising its thiol group to combat free radicals. This study observed that the GSH/GSSG ratio increased in all *C. papaya* treatments (Figure 6B). GSH is an antioxidant that donates hydrogen to ROS, converting itself into oxidised glutathione (GSSG) [68]. Consequently, the GSH/GSSG ratio serves as a biomarker for the body’s overall redox state. Hence, the elevation in GSH/GSSG in *C. papaya* treatments suggests that more GSH transformed into GSSG due to the elimination of hydrogen peroxide. Meanwhile, metformin reduced the GSH/GSSG ratio (Figure 6B), possibly due to its role in detoxifying ROS, similar to the effect observed in both *C. papaya* leaf and root, except for HGR1000. Furthermore, glutathione peroxidase-1 (GPx1), an enzyme dependent on GSH, converts hydrogen peroxide into water and oxygen [68]. All the *C. papaya* treatments increased GPx1 activity more than the HGC (Figure 7C), indicating the removal of hydrogen peroxide to protect the cells from ROS, as observed through the MDA levels (Figure 5A). Based on these antioxidant results, it appears that the leaf *C. papaya* extract effectively responded to ROS by increasing GSH (Figure 6A), and GPx1 (Figure 7C), as SOD2 was used up while detoxifying superoxide (Figure 7B) and ultimately keeping ROS levels at a minimum (Figure 5A). However, cells treated with HGR1000 exhibited a significant increase in MDA levels (Figure 5A) attributed to elevated ∆Ψm (Figure 4A), potentially leading to the formation of RNS due to superoxide leakage during the ETC, reacting to NO [69].

The NO levels can worsen oxidative stress by contributing to the formation of RNS, which in turn leads to more cellular damage and inflammation [69]. This study showed that all the *C. papaya* extracts elevated NO levels in HepG2 cells, potentially increasing the production of RNS and contributing to oxidative stress (Figure 5B). However, HGMet decreased NO levels (Figure 5B), indicating a possible protective effect against oxidative stress. The NO is essential for various physiological processes like vasodilation, neurotransmission, inflammation, and immune responses. Maintaining its bioavailability is crucial [70]. Still, under HG conditions, the superoxide radical’s production rapidly “inactivates” NO, forming peroxynitrite, a potent oxidant. However, there was no significant change in free radicals-related damage in *C. papaya*-treated cells as indicated by moderate levels of MDA, except in HGR1000 (Figure 5B). The production of NO is closely linked to the expression of inducible nitric oxide synthase (iNOS), which synthesises NO from L-arginine, O_2_, and NADPH [71]. The *C. papaya* treatment significantly increased iNOS protein expression compared to the HGC, similar to the effect of metformin (Figure 7A). This higher expression of iNOS might have contributed to the subsequent increase in NO levels (Figure 5B). Moreover, iNOS efficiently regulates glucose metabolism during hyperglycemia-induced inflammation [72], potentially explaining the increased levels observed in this study (Figure 7A).

The excessive accumulation of ROS and RNS further stimulates the ARE-driven genes, regulated by Nrf2, to transcribe the synthesis of antioxidants [14]. This study showed that both the leaf extract of *C. papaya* and metformin upregulated Nrf2 gene expression in cells exposed to HGC, with a significant increase observed in the *C. papaya* leaf-treated cells (Figure 8A). However, the Nrf2 protein expression decreased non-significantly in HepG2 cells treated with HGL500 and HGL1000 (Figure 8B). Conversely, this was not observed in metformin as it was significantly decreased. The non-significant reduction of protein expression in *C. papaya* leaf demonstrates a responsive role to free radicals induced by hyperglycemia that is achieved through the upregulation of the *Nrf2* gene (Figure 8A). The binding of Nrf2 to ARE increases the availability of SOD2, GPx1 and GSH antioxidants, effectively preventing oxidative damage (Figure 5A). Since the *NFE2L2* gene also contains ARE within its promoter region, *Nrf2* gene expression is upregulated to respond to the depleted Nrf2 protein levels [73]. Similarly, metformin slightly induces the Nrf2 gene response above the HCG (Figure 7B), leading to the upregulation of GSH (Figure 6A), GPx1 (Figure 7C), and SOD2 (Figure 7B) to protect cells from ROS damage (Figure 5A), despite the reduction in protein expression (Figure 8B). However, in *C. papaya* root extract, particularly in HGR1000-treated cells, both ROS and RNS were increased (Figure 5A,B), which could have stimulated the GSH (Figure 5A) and GPx1 (Figure 7C) antioxidants defence. However, the *Nrf2* gene and protein expression were significantly decreased (Figure 8), implying the possible compromise of cells from oxidative damage. Even so, the HGR500 significantly increased Nrf2 protein expression, hence, the antioxidants; Gpx1, GSH, and SOD2 were kept above the HGC, thus maintaining the MDA levels below the HCG (Figure 5A). Furthermore, various studies have highlighted the ability of *C. papaya* to counteract oxidative stress in hyperglycaemic conditions by reducing ROS and stimulating antioxidants [31,74,75,76]. Similarly, *C. papaya* seed extracts exhibited antioxidant activity against H_2_O_2_-induced oxidative stress in HepG2 cells by increasing levels of CAT, GSH, and GPx [75].

## 5. Conclusions

Based on the collected evidence, it can be concluded that *C. papaya* leaf and root extracts promote glucose uptake under HG conditions and upregulate antioxidants to protect cells from potential damage caused by free radicals from metabolism. The leaf extract successfully stimulated the metabolism of glucose while maintaining the normal function of mitochondria. However, it also triggered the production of ROS, leading to a strong response from the extract to stimulate the antioxidant GSH, GPx1, and Nrf2 gene expression. On the other hand, the root extract, especially at high concentrations, may not have been effective in detoxifying the elevated free radicals as the leaf extract because defence mechanisms, including the Nrf2 gene and protein expression, were reduced. However, SOD2, GSH, and GPx1 were kept above normal levels. Further studies should focus on understanding the mechanism by which glucose is taken into cells and exploring potential uses or storage. Additionally, in vivo, animal models should be utilised in future research to elucidate the actual mechanism of action.

## Figures and Tables

**Figure 1 nutrients-16-03496-f001:**
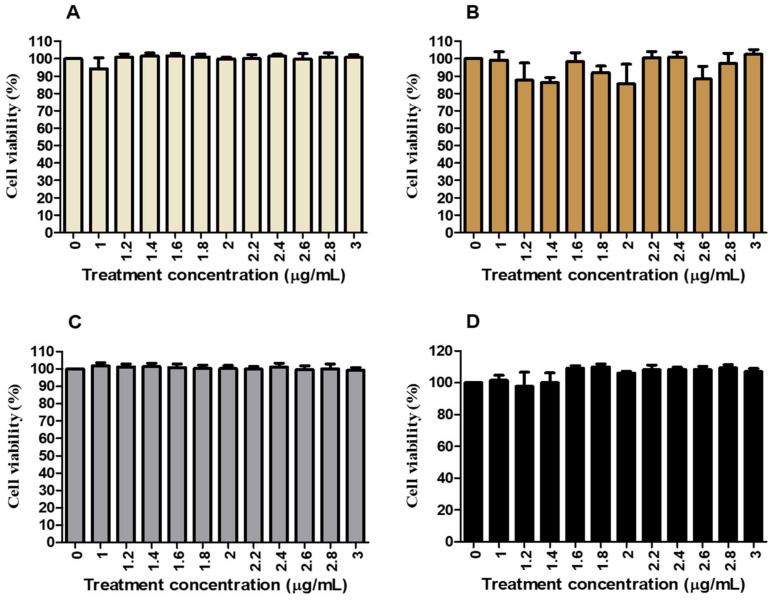
The effects of *C. papaya* leaf and root extracts on cell viability and cytotoxicity in HepG2 cells induced with NG and hyperglycaemia. *C. papaya* leaf extract had fluctuating effects on cell viability, ranging from 85% to 103%, in both NG and hyperglycaemia (HG) treated cells (**A**,**B**), respectively. However, cell viability remained similar to the control (100%) in NG cells treated with *C. papaya* root (**C**). In contrast, the viability was increased to 110% in HG-treated cells (**D**). All experiments were conducted in triplicates.

**Figure 2 nutrients-16-03496-f002:**
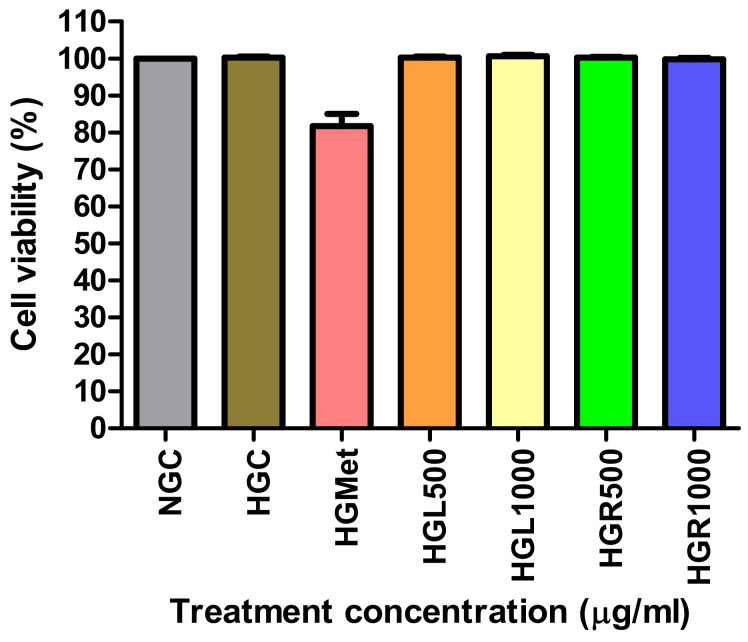
In hyperglycemic conditions, the concentration of *C. papaya* leaf and root extracts at 500 and 1000 µg/mL maintained cell viability similar to normal and HG controls. Metformin slightly reduced it to 81%.

**Figure 3 nutrients-16-03496-f003:**
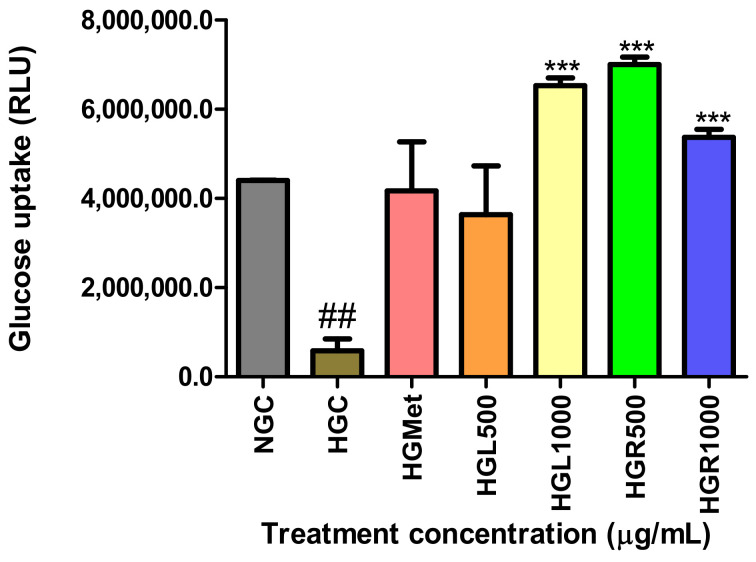
The levels of glucose in HGMet- and HGL500-treated cells were non-significantly decreased in relation to the HGC-treated cells, while were increased significantly in the HGL1000-, HGR500-, and HGR1000-treated cells (##, *** *p* < 0.05, unpaired student *t*-test with Welch’s correction).

**Figure 4 nutrients-16-03496-f004:**
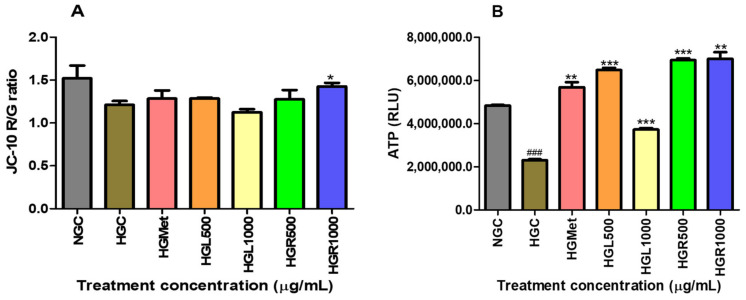
The effect of *C. papaya* on hyperglycaemia-induced HepG2 was shown through two measures: mitochondrial membrane potential and ATP activity levels. *C. papaya* treatment showed a slight increase in ∆Ψm except for HGR1000, where a significant increase was observed (**A**). Also, ATP levels in HG-treated HepG2 cells were significantly higher compared to HG-control (**B**) (###, *, **, ***, *p* < 0.05, unpaired student *t*-test with Welch’s correction).

**Figure 5 nutrients-16-03496-f005:**
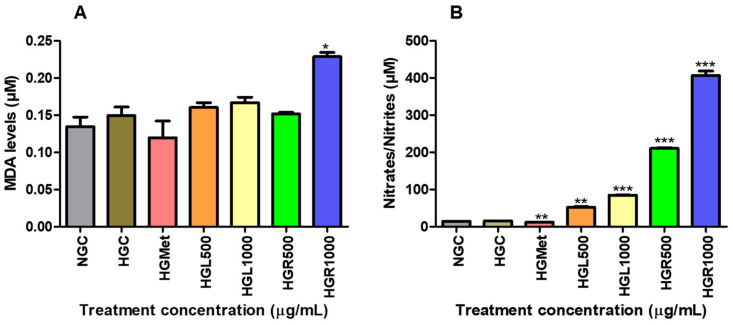
The effect of *C. papaya* leaf and root extracts on lipid peroxidation induced by HG (**A**) and nitrate/nitrite concentrations (**B**). Results showed that HGMet treatment led to a decrease in MDA levels, while *C. papaya* treatment resulted in an increased MDA concentration (**A**). Moreover, HGMet treatment significantly lowered nitrate/nitrile levels in cells, whereas *C. papaya* treatment led to a significant increase in nitrate/nitrile levels in HepG2 cells (**B**) (*, **, *** *p* < 0.05, unpaired student *t*-test with Welch’s correction).

**Figure 6 nutrients-16-03496-f006:**
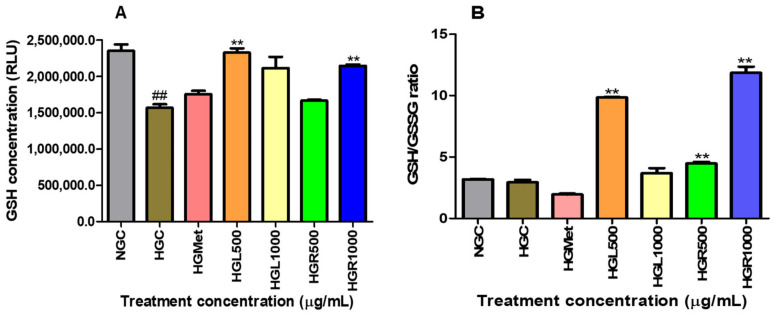
The presence of hyperglycaemia significantly reduced GSH levels from normal glucose to HGC. However, treatments effectively restored intracellular GSH levels above the HGC (**A**). The GSH/GSSG ratio decreased in HGMet-treated cells. At the same time, it increased in HG-induced cells treated with *C. papaya* leaf and root extracts, indicating the antioxidant effect of *C. papaya* extracts on GSH/GSSG in HG HepG2 cells (**B**) (##, ** *p* < 0.05, unpaired student *t*-test with Welch’s correction).

**Figure 7 nutrients-16-03496-f007:**
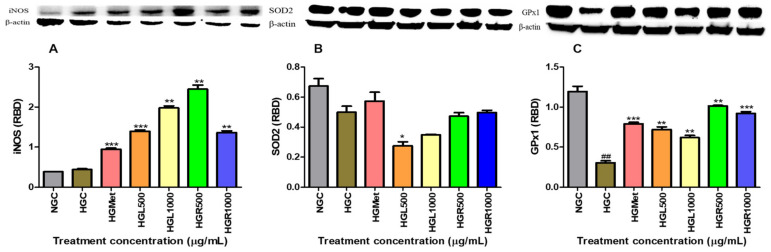
The effects of *C. papaya* on the iNOS, SOD2, and GPx1 protein expression of hyperglycaemia-induced HepG2 cells (**A**) shows a non-significant increase in iNOS protein expression from the NG-control to the HG-control; however, a significant increase in the protein expression was seen in HGMet and *C. papaya*-treated cells compared to the HG-control. (**B**) SOD2 expression was upregulated in HG cells treated with metformin but downregulated in *C. papaya* HGL500 and HGL1000. SOD2 expression remained slightly higher in HGR-treated cells compared to HGC. (**C**) GPx1 protein expression decreased in HG-control when compared to the NG-control, while it was increased in HGMet and *C. papaya*-treated HepG2 (##, *, **, *** *p* < 0.05, unpaired student *t*-test with Welch’s correction).

**Figure 8 nutrients-16-03496-f008:**
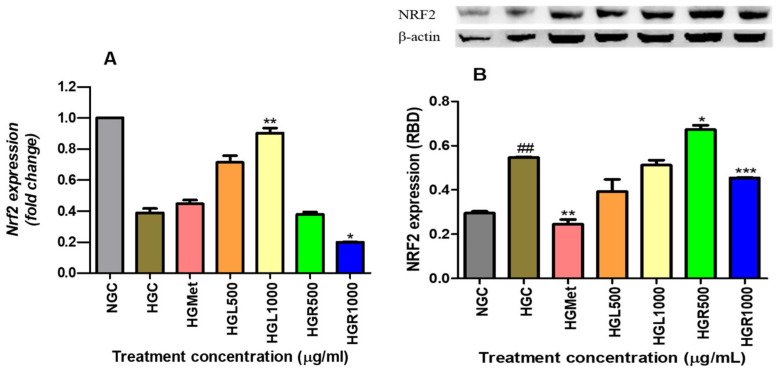
The gene expression of *Nrf2* was found to be lower in HGC compared to NGC, while its protein expression was significantly higher in HGC. However, when metformin, HGL500, and HGL1000 were administered, there was an increase in the gene expression of *Nrf2*, while the protein expression decreased as a result of the same treatments (**A**). On the other hand, treating hyperglycemic HepG2 cells with *C. papaya* root extract led to a downregulation of the Nrf2 gene expression, while HGR500 increased the protein expression. However, the protein expression was significantly reduced in cells treated with HGR1000 (**B**) (##, *, **, *** *p* < 0.05, unpaired student *t*-test with Welch’s correction).

**Table 1 nutrients-16-03496-t001:** Genes of Interest with Their Corresponding Primer Sequences and Annealing Temperatures.

Gene	Primers	Annealing Temperature (°C)
*CAT*	Forward: 5′-TAAGACTGACCAGGGCATC-3′ Reverse: 5′-CAACCTTGGTGAGATCGAA-3′	54.3
*GAPDH*	Forward: 5′-TCCCTGAGCTGAACGGGAAG-3′ Reverse: 5′-GGAGGAGTGGGTGTCGCTGT-3′	52.6
*NRF2*	Forward: 5′-AGTGGATCTGCCAACTACTC-3′ Reverse: 5′-CATCTACAAACGGGAATGTCTG-3′	54.3

**Table 2 nutrients-16-03496-t002:** Antibodies, catalogue numbers and dilutions.

	Antibody	Catalogue Number	Dilution
Primary Antibodies	SOD2 (D3X8F) XP® Rabbit mAb	13141 (Cell signalling technology)	1:1000 in 2% BSA
	iNOS (D6B6S) Rabbit mAb	13120 (Cell signalling technology)	1:1000 in 2% BSA
	GPx1 Rabbit mAb	3286 (Cell signalling technology)	1:1000 in 2% BSA
	Rabbit Anti-Nrf2	ab137550 (Abcam; thermo fisher scientific, SA, Jhb)	1:1000 in 2% BSA
Secondary Antibody	Anti-rabbit IgG, HRP-linked Antibody	7074 (Cell signalling technology)	1:1000 in 2% BSA
Housekeeping antibody	Anti-β-actin	A0bD12141 (Sigma-Aldrich; Merck)	1:5000 in 2% BSA

**Table 3 nutrients-16-03496-t003:** The Glucometer Kit Was Used to Determine the Concentrations of the Glucose Uptake Post-*C. papaya* Treatment and Metformin.

Treatment (µg/mL)	Glucose Levels (mmol/L)
NGC	Low
HGC	7.8
HGMet	Low
HGL500	Low
HGL1000	Low
HGR500	Low
HGR1000	Low

Low: below detectable levels.

## Data Availability

The generated data used to support the findings of this study are included within the article.

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
