# Peer review of "Mitigating Hyperglycaemic Oxidative Stress in HepG2 Cells: The Role of Carica papaya Leaf and Root Extracts in Promoting Glucose Uptake and Antioxidant Defence"

_nutrients, 2024, doi:10.3390/nu16203496_

Round 1

Reviewer 1 Report

Comments and Suggestions for Authors

The authors show very interesting properties of Carica papaya leaf and root extracts in promoting glucose uptake and antioxidant defence on HepG2 and Hek293 cells.

*Minor corrections:

RNS is reactive nitrogen species, please correct line 65-66

2.4. Treatment of cell please writte Treatment of cells

Please writte µL, mL etc instead of µl, mL etc (to be corrected in the whole text, Tables AND Figures)

*Major corrections:

Introduction; The aim of the study must be better explained line100-102

2.7.2. Mitochondrial Membrane Potential (ΔΨm) Assay; please explain how works JC-10 dye (refs required)

2.8. Spectrophotometry assays; additionnal details required, not clear

2.10. Western Blotting, the references of the antibodies used must be clearly indicated. Very confusing under the present form.

2.11. Data analysis : OK, appropriated statistics

Results

As 2 cell lines have been used, can new indicate the name of the cell used in the different legends. Mandatory.

Figure 3.8: the data of Western Blot are of bad quality and must be improved. Better presentations are required. Whole western blot must be provided in Supplementary materials.

Discussion OK

Conclusions, can be improved: interest of the data and applications (human animals).

Author Response

The authors show very interesting properties of Carica papaya leaf and root extracts in promoting glucose uptake and antioxidant defence on HepG2 and Hek293 cells.

We would like to express our gratitude to the reviewer. Thank you for your advice, suggestions, and corrections on this manuscript. We appreciate the way you identified both minor and major errors and provided helpful, wise, and thoughtful suggestions and comments that have improved the quality of this manuscript. Please find below how we have addressed your comments, corrections, and suggestions.

*Minor corrections:

Comment 1: RNS is reactive nitrogen species, please correct line 65-66

Response one: This was corrected from nitrogen reactive species to reactive nitrogen species (line 74).

Comment 2: 2.4. Treatment of cell please write Treatment of cells

Response 2: This was corrected as suggested (line 150)

Comment 3: Please writte µL, mL etc instead of µl, mL etc (to be corrected in the whole text, Tables AND Figures).

Response 3: Please remember that the units were written as "µL" and "mL" throughout the entire text.

*Major corrections:

Comment 4: Introduction; The aim of the study must be better explained line100-102

Response 4: The aim of the study was improved to include “investigating the effects of extracts on glucose uptake and cell response to hyperglycemic-induced oxidative stress, compared to metformin” (lines 118-121).

Comment 5: 2.7.2. Mitochondrial Membrane Potential (ΔΨm) Assay; please explain how works JC-10 dye (refs required).

Response 5: The information about how the JC-10 works and why it is used to quantify the ΔΨm was added as per a suggestion (see line 224-228).

Comment 6: 2.8. Spectrophotometry assays; additional details required, not clear

Response 6: The additional information on the preparation of cells and treatment medium was included as recommended (line 248-254).

Comment 7: 2.10. Western Blotting, the references of the antibodies used must be clearly indicated. Very confusing under the present form.

Response 7: The western blotting antibodies, catalogue numbers, and dilutions were updated and presented in a table (see Table 3).

2.11. Data analysis: OK, appropriated statistics

Results

Comment 8: As 2 cell lines have been used, can new indicate the name of the cell used in the different legends. Mandatory.

Response 8: The Hek293 cell line was only used for the cytotoxicity assay as shown in Figure 2. However, only the HepG2 cell line was used for the subsequent assays.

Comment 9: Figure 3.8: the data of Western Blot are of bad quality and must be improved. Better presentations are required. Whole western blot must be provided in Supplementary materials.

Response 9: The quality of the Western Blot image bands has been enhanced, and additional original Western Blot images have been provided in Supplementary materials.

Discussion OK

Comment 10: Conclusions, can be improved: interest of the data and applications (human animals).

Response 10: The conclusion has been revised, and necessary improvements have been made.

Reviewer 2 Report

Comments and Suggestions for Authors

Diabetes, and especially type II diabetes, is one of the most common diseases in the world and although it is not fatal, the effects it causes are among the most common causes of death among people. This problem concerns not only highly developed countries but also medium-developed and developing ones. WHO forecasts regarding the increase in the number of cases in the coming years are not optimistic, which means that many research centers are looking for both new drugs that can support glucose control in people with diabetes and substances that can support their diet.

The introduction to the article was written correctly based on well-chosen current literature. The results obtained by the authors were obtained using standard methods, but the way they were described leaves much to be desired. The authors showed little care in their description, making a number of errors:

1.      As a rule, units should be separated by a space from the numerical value. Authors sometimes wrote them without a space (lines: 122, 141, 145 and many others) and sometimes with a space (lines: 279, 281, 283 and many others).

2.      Authors should standardize volume units, e.g.: line 116 - cm3, line 144 - ml, line 145 μl, line 223 - μL.

3.      The Celsius degrees should also be improved and unified, see e.g. lines 147, 155, 340.

4.      Line 170 should be 2-deoxy-D-glucose, and line 171 should be 2-deoxy-D-glucose-6-phosphate.

5.      Line 174 should be CO2.

6.      Line 277 - why Tables 3.1 and not Table 1. By default, tables are numbered 1, 2, 3 ....

7.      Line 371 - Shouldn't B be D?

8.      Line 373 - Should be Figure 1.

9.      The text contains acronyms for which no explanation is provided, e.g. NGC.

-.     Line 397 - Which table does the designation Table 1 refer to?

   Line 398 - Next Table 3.1.

1     Line 436 - C. papaya should be written in italics.

1    Line 595 - The letter D in the name of sugar should be written in a font 2 points smaller than the rest.

1    References - literature entries should be supplemented with DOI numbers.

The errors listed above are just a few of many in the text. Authors should carefully proofread the article before resubmitting it.

Author Response

Reviewer 1

Diabetes, and especially type II diabetes, is one of the most common diseases in the world and although it is not fatal, the effects it causes are among the most common causes of death among people. This problem concerns not only highly developed countries but also medium-developed and developing ones. WHO forecasts regarding the increase in the number of cases in the coming years are not optimistic, which means that many research centers are looking for both new drugs that can support glucose control in people with diabetes and substances that can support their diet.

The introduction to the article was written correctly based on well-chosen current literature. The results obtained by the authors were obtained using standard methods, but the way they were described leaves much to be desired. The authors showed little care in their description, making a number of errors:

General response: Thank you for taking the time to thoroughly review our manuscript. Your comments and recommendations demonstrate your passion and investment in evaluating our manuscript. Your attention to detail is evident in the minor typographical and grammatical errors you identified throughout the manuscript. We feel honoured to have my manuscript reviewed by you. The comments and recommendations were addressed as follows.

Comment 1. As a rule, units should be separated by a space from the numerical value. Authors sometimes wrote them without a space (lines: 122, 141, 145 and many others) and sometimes with a space (lines: 279, 281, 283 and many others).

Response 1: The units were separated by a space from the numerical value throughout the manuscript (e.g. lines: 279, 281, 283)

Comment 2. Authors should standardize volume units, e.g.: line 116 - cm3, line 144 - ml, line 145 μl, and line 223 - μL.

Response 2: The volume units were standardized to mL and were used throughout.

Comment 3. The Celsius degrees should also be improved and unified, see e.g. lines 147, 155, 340.

Response 3: The Celsius degrees were improved and unified see e.g. lines 147, 155, 340.

Comment 4. Line 170 should be 2-deoxy-D-glucose, and line 171 should be 2-deoxy-D-glucose-6-phosphate.

Response 4: The missing letter D in lines 170, and 171 was added to 2-deoxy-D-glucose to 2-deoxy-D-glucose-6-phosphate.

Comment 5. Line 174 should be CO2.

Response 5: CO2 was corrected

  1. Line 277 - why Table 3.1 and not Table 1. By default, tables are numbered 1, 2, 3 ....

Response 6: The table in line 277 was corrected to Table 1, and the subsequent table was also corrected.

Comment 7. Line 371 - Shouldn't B be D?

Response 7: Yes. It was supposed to be. and this was corrected.

Comment 8. Line 373 - Should be Figure 1.

Response 8: Figure 3.1 in line 373 was changed to Figure 1 and all other Figures were corrected. 

Comment 9. The text contains acronyms for which no explanation is provided, e.g. NGC.

Response 9: The acronyms were previously spelt out in full in the methodology and were also included in the subsequent text. e.g. normal glucose control (NGC) and high glucose control (HGC) in lines 397 and 398.

Comment 10. Line 397 - Which table does the designation Table 1 refer to?

Response 10: This was referring to table 3.1 which has been corrected to table

Comment 11: Line 398 - Next Table 3.1.

Response 11: This was meant to be table 4.1 which has been corrected to table 2.

Comment 12. Line 436 - C. papaya should be written in italics.

Response 12: C. papaya in line 436 was italicized and this was checked throughout the entire manuscript.

Comment 13. Line 595 - The letter D in the name of sugar should be written in a font 2 points smaller than the rest.

Response 13: The letter D in 2-deoxy-D-glucose and 2-deoxy-D-glucose-6-phosphate was written in a font 2 points smaller than the rest. See lines 186, 187, 193, 632 and 633.

References - literature entries should be supplemented with DOI numbers.

Response: The DOI numbers were included in the references.

The errors listed above are just a few of many in the text. Authors should carefully proofread the article before resubmitting it.

Response: The authors appreciate the reviewer's feedback and have made necessary corrections and thorough proofreading before resubmission.

Reviewer 3 Report

Comments and Suggestions for Authors

 In this study authors tested cytotoxicity, evaluated glucose uptake and metabolism, and examined antioxidant properties in HepG2 cells treated with C. papaya aqueous leaf and root extract. Authors found that C. papaya extracts did not exhibit toxicity in HepG2 cells and enhanced glucose uptake compared to the hyperglycaemic control (HGC) and metformin. Moreover, C. papaya triggered an antioxidant response upregulating GPx1, GSH, catalase, and Nrf2, ultimately lowering ROS. The root extract stimulated SOD2, GPx1, and GSH levels, reducing catalase and Nrf2 gene expression.

Although the manuscript is interesting, it presents several points that need improvements.

Introduction: Although authors investigated the expression of NRF2-dependent genes and NRF2 itself, this transcription factor is not evene mentioned in the introduction. However, this transcription factor has a key role in several diseases (PMID: 37296999,PMID: 38812389) and deserves to be introduced.

Lines 34-36: It deserves to be pointed out that several of the pathologies mentioned are mainly due to the high glucose-induced endothelial dysfunction, which is a characteristic of diabetes mellitus (PMID: 37443812). 

Table 3 must be corrected in Table 1. Moreover, authors must add the primers sequences of housekeeping gene 

Please, add the number of replicates in the legends of the figures

Figures numbering must be correct

Figure 1: Statistical significant differences must be shown with asterisks

Authors must add the molecular weight in western blot images

Figure 8: Why authors did not investigate NRF2 and CAT protein expression by western blot? these data should be added to validate 

Lines 754-755: Why authors added an ethical approval if no human or animal tissue was used? 

What is the reason of using  Hek293 cells only in the first experiments if all the manuscript is focused on HepG2 cells?

Author Response

Reviewer 2

In this study, authors tested cytotoxicity, evaluated glucose uptake and metabolism, and examined antioxidant properties in HepG2 cells treated with C. papaya aqueous leaf and root extract. The authors found that C. papaya extracts did not exhibit toxicity in HepG2 cells and enhanced glucose uptake compared to the hyperglycaemic control (HGC) and metformin. Moreover, C. papaya triggered an antioxidant response upregulating GPx1, GSH, catalase, and Nrf2, ultimately lowering ROS. The root extract stimulated SOD2, GPx1, and GSH levels, reducing catalase and Nrf2 gene expression.

Although the manuscript is interesting, it presents several points that need improvements.

General response: We want to express our gratitude to the reviewer for thoroughly examining our article. The comments and suggestions have significantly enhanced our manuscript. We feel honoured that our work was reviewed by you.

Comment 1: Introduction: Although the authors investigated the expression of NRF2-dependent genes and NRF2 itself, this transcription factor is not even mentioned in the introduction. However, this transcription factor has a key role in several diseases (PMID: 37296999,PMID: 38812389) and deserves to be introduced.

Response 1: Thank you for bringing this to our attention. We have included information about NRF2, its regulation, and the transcriptional role of antioxidants (see lines 78-84).

Comment 2. Lines 34-36: It deserves to be pointed out that several of the pathologies mentioned are mainly due to the high glucose-induced endothelial dysfunction, which is a characteristic of diabetes mellitus (PMID: 37443812). 

Response 2: Thank you for the suggestion. It has been reviewed and added as suggested (refer to lines 93-94).

Comment 3. Table 3 must be corrected in Table 1. Moreover, the authors must add the primers sequence of the housekeeping gene 

Response 3: Table 3.1 was corrected to Table 1 and GAPDH primers were added.

Comment 3. Please, add the number of replicates in the legends of the figures

Response 3: This was added as suggested.

Comment 4. Figures numbering must be correct

Response 4 : Figure numbers were corrected from Figure 1 to Figure 8

Comment 5. Figure 1: Statistically significant differences must be shown with asterisks

Response 5: There was no significant difference in cell viability induced by the different concentrations.

Comment 6. Authors must add the molecular weight in western blot images

Response 6: The BLUeye Prestained Protein Ladder, a molecular weight marker, was used and run alongside the treatment samples. However, after the transfer, the marker only appeared on the membrane. Occasionally, it showed up once before stripping the membrane for the next protein. However, the molecular weight marker from the membrane and the guide were used to verify the correct protein.

Comment 7. Figure 8: Why did authors not investigate NRF2 and CAT protein expression by western blot? these data should be added to validate 

Response 7: Initially, the authors planned to include NRF2 and CAT in the study, but only the primers for CAT were available at that time. Now, the data for NRF2 is available and has been included in the articles. However, the data for catalase is still not available, so the authors have decided to remove catalase from the main text and supplement it in the appendix.

Comment 8. Lines 754-755: Why authors added an ethical approval if no human or animal tissue was used?

Response 8: ethics is a requirement that all studies at the University of KwaZulu-Natal (UKZN) must be submitted for BREC approval.

Comment 8. What is the reason of using  Hek293 cells only in the first experiments if all the manuscript is focused on HepG2 cells?

Response 9: The HEK293 cells were included to confirm that the extracts were not toxic, even to normal cells. However, if necessary, we can omit this.

Round 2

Reviewer 3 Report

Comments and Suggestions for Authors

manuscript can be accepted in the current form